Changes in the core species of the ant-plant network of oak forest converted to grassland: replacement of its ant functional groups

Cuautle Mariana mariana.cuautle@udlap.mx 1
Díaz-Castelazo Cecilia 2
Castillo-Guevara Citlalli 3
Torres Lagunes Carolina Guadalupe 1
1 Ciencias Químico Biológicas, Universidad de las Américas Puebla , Puebla , México
2 Red de Interacciones Multitróficas, Instituto de Ecología, A. C. , Xalapa , Veracruz , México
3 Centro de Investigación en Ciencias Biológicas, Universidad Autónoma de Tlaxcala , San Felipe Ixtacuixtla , Tlaxcala , México
Rana Naureen
Electronic publication date: 2022 Jul 13
Publication date: 2022
Volume: 10
Electronic Location ID: e13679
Received 2022 Jan 11; Accepted 2022 Jun 13
Copyright: ©2022 Cuautle et al.
Copyright year: 2022
Copyright holder: Cuautle et al.
License: This is an open access article distributed under the terms of the Creative Commons Attribution License, which permits unrestricted use, distribution, reproduction and adaptation in any medium and for any purpose provided that it is properly attributed. For attribution, the original author(s), title, publication source (PeerJ) and either DOI or URL of the article must be cited.
License URL: https://creativecommons.org/licenses/by/4.0/

Keywords: Land-use change, Mexico, Nested network, Ant-plant interactions

Funding: Consejo Nacional de Ciencia yTecnología 223033 The Consejo Nacional de Ciencia yTecnología (CONACYT: 223033) provided Mariana Cuautle with financial support. The funders had no role in study design, data collection and analysis, decision to publish, or preparation of the manuscript.

==============================
Land-use change in terrestrial environments is one of the main threats to biodiversity. The study of ant-plant networks has increased our knowledge of the diversity of interactions and structure of these communities; however, little is known about how land-use change affects ant-plant networks. Here we determine whether the change in land use, from native oak forest to induced grassland, affected the network properties of ant-plant networks in a temperate forest in Mexico. We hypothesize that the disturbed vegetation will be more nested and generalized due to the addition of generalist species to the network. The oak forest network comprises 47 plant species and 11 ant species, while the induced grassland network has 35 and 13, respectively. Floral nectar was the resource used most intensely by the ants in both vegetation types. The ant-plant network of the induced grassland was significantly more nested and generalist than that of the oak forest; however, none of the networks were nested when considering the frequency of interaction. In both vegetation types, the ants were more specialized than the plants, and niche overlap was low. This could be related to the dominant species present in each type of vegetation: Prenolepis imparis in the oak forest and Camponotus rubrithorax in the grassland. The central core of cold climate ant species in the oak forest was replaced by a central core of subordinate Camponotini and tropical specialists in the induced grassland. These results suggest that the increase in nestedness and generalization in the grassland may be related to the loss of the cold climate specialists from the core of the oak forest network. Our findings provide evidence that land-use change increases the level of generalization in the ant-plant interaction networks of temperate forests.

Indroduction

The loss of biodiversity is considered one of humanity’s main problems (Ceballos et al., 2015). Conserving biodiversity is vital to human well-being as species extinctions have great potential to alter the properties (Hooper et al., 2005) that regulate practically all the different ecosystem services we depend on (Balvanera et al., 2009; Majeed et al., 2020; Ramzan et al., 2021). Therefore, due to anthropogenic activities, we are now facing a sixth mass extinction (Ceballos et al., 2015). The main causative factors of this biodiversity loss are the introduction of invasive species, climate change, changes in atmospheric CO2 concentration, biogeochemical cycles, and land-use change (Sala et al., 2000; Majeed et al., 2020; Ramzan et al., 2021). In terrestrial ecosystems, land-use change (i.e., anthropogenic alteration of natural vegetation; Underwood & Fisher, 2006; Del Toro, Ribbons & Pelini, 2012) has been cited as the most important determinant of biodiversity loss (Sala et al., 2000). Biodiversity is a concept that encompasses several aspects, including the relative abundance of species, gene variance within species, and the variety of communities and ecosystems in which they live (Balvanera et al., 2009). Although it is not always considered, biodiversity also includes the diversity of interspecific interactions (Tylianakis et al., 2010). Interspecific interactions have determined the evolution and diversification of species (Thompson, 1982). The history of evolution and biodiversity is essentially a history of the evolution of interactions between species (Thompson, 1982).

Complex interaction network analysis has made it possible to study different types of interspecific interactions (e.g., plant-pollinator, ant-plant, plant-frugivore, host-parasitoid, host-parasite) at the level of entire communities and to detect their interaction patterns (Bascompte & Jordano, 2007; Dehling, 2018). In the analysis of complex interaction networks, species are represented as nodes, and the interaction between different species is shown as links connecting them (Bascompte & Jordano, 2007). Studies of different of network types reveal similarities and fundamental differences, e.g., mutualistic networks tend to be nested, and antagonistic networks tend to be compartmentalized (Bascompte, 2010; Thébault & Fontaine, 2010). Nested networks are characterized by specialist species interacting with subgroups of the species with which generalists interact. Their nested structure appears to contribute in community resilience (i.e., they are very robust to the loss of most species but very fragile to the extinction of the most generalist species, Bascompte & Jordano, 2007). Complex interaction networks have increased our understanding of the relationship between complexity and stability, the non-random nature of networks, and how this complexity is configured (Ings & Hawes, 2018). The study of ant-plant networks has contributed to our general knowledge of network structure (Del-Claro et al., 2018). The most studied ant-plant networks are those between plants with extrafloral nectaries (i.e., sugar secretory glands on the plants not associated with pollination; Elias, 1983) and facultative ants (i.e., ants that use the nectar resource offered by a plant but are not restricted to that particular plant species, see Rico-Gray, 1993). In this type of interaction, the plant offers extrafloral nectar as a reward for the ants, and in return, the ants defend the plant against herbivores (Rico-Gray & Oliveira, 2007). It is important to mention that studies evaluating the network structure of this type of interaction have just assumed the interaction was mutualistic (e.g.,  Guimarães et al., 2006; Guimarães Jr et al., 2007; Díaz-Castelazo et al., 2010; Dáttilo, Díaz-Castelazo & Rico-Gray, 2014). Confirmation of the nature of these interactions would have required that an experiment be performed (e.g.,  Rico-Gray & Thien, 1989; Oliveira et al., 1999; Cuautle & Rico-Gray, 2003). Regardless, few studies have evaluated the network structure of ant-plant interactions in which the plant offers other resources besides extrafloral nectar (honeydew, floral nectar, fleshy fruits, nesting sites) (but see Corro et al., 2019). Potential outcomes of these interactions are commensalistic (the ants benefit by foraging on the plant), mutualistic (the plant is defended, pollinated or its seeds dispersed by the foraging ants), antagonistic (the ants are nectar robbers, seed predators or herbivorous) or even neutral (the ants just use the plants as substrates for foraging and patrolling) (Rico-Gray & Oliveira, 2007; Corro et al., 2019). This work evaluates ant-plant networks in which different types of resources, especially sugars, are foraged. We explore how the structure of these networks is affected by disturbance (e.g., land-use change).

Ant-plant networks have proven to be robust, i.e., maintaining their nesting pattern despite different types of disturbance (Passmore et al., 2012; Sánchez-Galván, Díaz-Castelazo & Rico-Gray, 2012; but see Lara et al., 2020). However, disturbances can cause alterations, such as the loss of specialists (Passmore et al., 2012), changes in the roles of species in the community (Sánchez-Galván, Díaz-Castelazo & Rico-Gray, 2012; Lara et al., 2020; Câmara et al., 2018), or the level of specialization of their interactions (Falcão, Dáttilo & Izzo, 2015; Corro et al., 2019). These changes may have detrimental effects on the community. They can even cause it to disassemble, resulting in less-connected groups of species, leading to a random (Lara et al., 2020) or compartmentalized structure. When communities fragment into disjointed groups of species, they are more susceptible to species loss (Bascompte & Jordano, 2007). In addition to nestedness, different types of indices serve to characterize a network (Antoniazzi Jr, Dáttilo & Rico-Gray, 2018). These indices primarily address the question of how specialized or generalized the network is, its trophic levels, or the species that comprise it. All organisms specialize (Thompson, 1982), but the degree to which they specialize has always been a question of interest to ecology and evolution (Guimarães et al., 2006; Blüthgen et al., 2007; Schleuning et al., 2012). Disturbance has been shown to reduce the degree of specialization of ant-plant networks; at the species level (Passmore et al., 2012; Emer, Venticinque & Fonseca, 2013), trophic level (Câmara et al., 2018), and network level (Emer, Venticinque & Fonseca, 2013; Corro et al., 2019; but see Sánchez-Galván, Díaz-Castelazo & Rico-Gray, 2012; Falcão, Dáttilo & Izzo, 2015; Alves-Silva1 et al., 2020. Additionally, the identity of the specialist species may change (Lara et al., 2020). These results indicate that specialist species may be more susceptible to disturbance.

Most studies of ant-plant interaction networks were carried out in tropical ecosystems (80% of the studies, Del-Claro et al., 2018) and in networks where the resource offered by the plants is extrafloral nectar (62% of the studies, Del-Claro et al., 2018). Therefore, little is known about how these networks are structured in temperate ecosystems, where other resources are offered in addition to extrafloral nectar (e.g., floral nectar and honeydew). Oliveira & Koptur (2017) review and discuss how plant-ant interactions respond to disturbance, mainly in the tropics. However, few studies evaluate how ant-plant networks in temperate environments respond to disturbance (but see Lara et al., 2020). Disturbance or fragmentation can facilitate the presence of generalists, opportunist and/or the loss of specialist species in plant-insect networks (Benítez-Malvido et al., 2014; Emer, Venticinque & Fonseca, 2013; McKey & Blatrix, 2017; Lara et al., 2020). Generalist species tend to have greater niche breadth, which is related to more nested interactions (Blüthgen, 2010; Benítez-Malvido et al., 2014). For example, habitat fragmentation due to the construction of a dam in Central Amazonia led to less compartmentalized ant-myrmecophyte plant networks due to the loss of specialist mutualistic ant species that were replaced by opportunistic, probably non-mutualistic, ant species (Emer, Venticinque & Fonseca, 2013).

The objective of this study was to determine if the change in land use, from native oak forest to secondary native induced grassland, affects the network properties in the facultative ant-plant networks of a temperate forest in Mexico. We hypothesize that both networks would be nested because facultative ants tend to be generalists in their use of resources, and this generalization can lead to a nested pattern. We hypothesize that disturbance (i.e., land-use change) would cause a loss of specialist species in the areas of disturbed vegetation, generating a more nested and generalistic network dominated by generalist/opportunistic ants than the original network (generalist species forming part of the core of the network). In the oak forest, we anticipate a higher specialization level (at the species, trophic, and network levels) than the disturbed network, as it is expected to be dominated by specialists. A number of network parameters were used to measure the level of generalization/specialization at the network (H2, level of specialization), trophic (SA, specialization asymmetry; NO, niche overlap) and species level (degree; d’, species specialization). Finally, the most important species (SS, species strength and PP, direction of the interaction asymmetry) and the central core species for each network were identified. The ants were classified into functional groups to identify the role of the central core species.

Materials and Methods

Description of the study site

The study was carried out in the “Flor del Bosque” State Park (FBSP, Figs. 1 and 2, see Table 1 for characteristics of the study site), Puebla, Mexico. The park has 699.2 ha and is located approximately 10 km southwest of the capital of the state of Puebla, in the municipality of Amozoc de Mota (19°01′N, 98°20′W, 2,225–2,400 masl). FBSP presents three types of vegetation: oak forest, secondary induced grassland, and abandoned plantations of Eucalyptus spp. (Badano et al., 2010). The oak forest covers 255.68 ha of the park’s total surface, that is, 41.71%; grassland constitutes 16.25%, and eucalyptus plantations cover 1.52% (Costes Quijano et al., 2006). The remaining 40.52% of the surface consists of thorny scrub and urban areas (Costes Quijano et al., 2006). Grasslands and Eucalyptus plantations form patches within the oak matrix (Cuautle per. Obs.). In the oak forest, the tree layer varies between 5 and 10 m, with thin-branched trunks (Costes Quijano et al., 2006). The most abundant oak species are Quercus castanea Née and Q. laurina Bonpl. There are also species such as Cupressus lusitanica Mill, Juniperus deppeana Steud, J. flaccida var. flaccida Schltdl, Juniperus poblana (Martínez) R. P. Adams. The most abundant shrub species are Salvia polystachya Ort., followed by Mimosa aculeaticarpa Ortega, Eysenhardtia polystachya (Ortega) Sarg., Barkleyanthus salicifolius (Kunth) H.Rob. & Brettell and Lantana velutina M. Martens & Galeotti (Badano et al., 2010). The grasslands have only a few isolated adult oak trees. The oaks in the pastures are relictual trees of the original forest that were left standing to provide shade for the cattle that grazed in this area. In the grasslands, all shrub species have low densities (Badano et al., 2010). The shrub strata in the grassland are represented by species such as Agave potatorum Zucc., Agave salmiana Otto ex Salm-Dyck, Opuntia pilifera F.A.C. Weber, Opuntia tomentosa Salm-Dyck, Arctostaphylos pungens Kunth, Acacia schaffneri (S. Watson) F.J. Herm, Calliandra grandiflora (L’Hér.) Benth., Mimosa acanthocarpa Poir. and Mimosa aculeaticarpa (Badano et al., 2010).

Figure 1 Study site, Flor del Bosque State Park.

Left: oak forest, Right: induced grassland. Photo taken by M. Cuautle.

Figure 2 Map of the study zone.

Flor del Bosque State Park (FBSP), situated in Puebla State, in Mexico.

We chose to analyze the effect of land-use change for this study, comparing the original oak forest to the grassland. The grassland is secondary, resulting from the massive felling of oak forest for grazing 80 years ago, and is where the forest has not recovered naturally (Costes Quijano et al., 2006; Badano et al., 2010). The area was decreed a recreational, ecological zone in 1985, and since 2004 grazing activities within the park have been interrupted. Therefore, at the time of the sampling, the grassland had not been subject to grazing for approximately 11 years. In this work, we refer to the grassland as induced to clarify that it was the result of anthropogenic activities.

Ant-plant interaction monitoring

The sampling was carried out in three periods. The first took place from March to December of 2015 (except May, September, and October), the second from January to December (except June) of 2016, and the third in April and July of 2017. These sampling periods were chosen to obtain data from both wet and dry seasons, as ant-plant interactions are affected by seasonal temperature and changes in precipitation (Rico-Gray et al., 1998). The omission of certain months was due to the park closing provisionally to all public or for logistical reasons. At each site (induced grassland and oak forest, Fig. 2), six transects were drawn and designated T2, T5, and T6 for the oak forest and T1, T3, and T4 for the induced grassland. Each transect measured 400 m, and all plants present within 10 m on each side of the entire length of the transect were considered. Walking censuses were undertaken from 8:00 a.m. to 4:00 p.m., on days with mild weather conditions, initiating from a different paired transect and a different vegetation type to avoid order effects and the confounding effect of detection of interactions in the field. All trees and shrubs in the transects were searched for ants from the ground up to an approximate height of 1.5 m., which was the maximal height we could observe during our walking census. However, we were able to evaluate ant activity on tree species when ants were seen on the branches or base of the tree trunks. During these observations, the plants and the associated ants were all recorded using their assigned code during these observations. The number of ants on each plant was counted, and what type of resource the ant was foraging (floral nectar, extrafloral nectar, or hemipteran honeydew). Sometimes it was not possible to identify a specific resource; however, the ant displayed foraging behavior (i.e., touching a part of the plant with its mouthparts for more than 30 s and moving its abdomen) when foraging different individuals of the same plant species.

Table 1 Characteristic of the study zone, obtained from Costes Quijano et al. (2006).

Characteristic	Condition	
Climate	Temperate-subhumid	
Temperature	14 to 16 °C	
Precipitation	750 to 950 mm	
Rainy season	In the summer	
Dry season	From November to April	
Soil type	Lithosol and cambisol	

Transect sampling was initiated at different hours of the day and was carried out by a single person; it lasted approximately one to two hours. All observers were trained by the same person (MC) to avoid bias. The plants and ants were collected for later mounting and identification. Tweezers or entomological vacuums were used to collect the ants manually. The ants were identified at the genus level using the Mackay & Mackay key (1989). Ants and plants were sent to specialists for their identification. Specimens of the ants are conserved in the UDLAP Formicidae collection. The Secretaria de Medio Ambiente y Recursos Naturales (SEMARNAT) (Secretary of Environment and Natural Resources) approved the collection permit SGPA/DGV/06901/15 for this study.

Ant functional groups

Functional groups respond to the need to identify groups of organisms that transcend taxonomic and biogeographic boundaries and that respond predictably to stress (e.g., low temperatures, nest site availability, food supply) and disturbance (biomass removal). Andersen (2000) proposed seven ant functional groups: Dolichoderinae, are aggressive competitive ants found in environments with low levels of stress and disturbance; Subordinate Componotini, are behaviorally submissive to Dominant Dolichoderinae; The Climate Specialist, are ant specialists in hot, tropical, or cold weather; Cryptic Species, are more diverse and abundant in forested habitats; Opportunistic species, are unspecialized, poorly competitive, and ruderal; Generalized Myrmicinae, are in competitive tension with Dominant Dolichoderinae and are ubiquitous throughout the warmer regions of the world; and Specialist predators, are predators of arthropods, and have little interaction with other ants. We analyzed functional group identity in the networks because, according to Andersen (2000), we should be expecting the presence of certain functional groups, especially in their core. These were the Cold Climate Specialists in the oak forest network, and other types of functional groups in the induced grassland network (e.g., Generalized Myrmicinae). The use of functional groups gives us additional information about the differences in ant roles and levels of specialization in the two networks.

Analysis of data

Ant-plant interaction matrices were created for each study site; a qualitative matrix (0 absence of interaction, 1 presence of interaction) and a quantitative matrix (interaction frequency, i.e., the number of times that ant species i was observed with plant j). The qualitative network was used to determine the nestedness value for each network; the rest of the indices were established with the quantitative matrix. Nestedness was evaluated with the ANINHADO software using the NODF estimator. The network is considered nested if the NODF value is higher than predicted by the ER model (presences are randomly assigned to any cell within the matrix) with 1,000 random simulations (Almeida-Neto et al., 2008). We also computed the weighed version of NODF for quantitative matrices. This index ranges from zero (no nestedness) to 100 (perfect nestedness) (Almeida-Neto et al., 2008; Almeida-Neto & Ulrich, 2011). The different specialization indices at the network and species levels and the indices that allow for the identification of key species in the network were calculated with the R-package bipartite (v3.5.3; Dormann et al., 2009). The following metrics were computed: (i) degree (D), the number of different species a certain species interacts with (i.e., the number of links a species has), also known as the generalization level (Bascompte & Jordano, 2007); (ii) the H2 index shows the level of specialization of the interaction network, it is expressed in values between 0 and 1 where the value 0 indicates that there is no specialization and the value 1 indicates that there is full specialization (Blüthgen, Menzel & Blüthgen, 2006); (iii) specialization asymmetry (SA), based on the degree of specialization of each species (d’), since the mean d-value for the lower trophic level (plants) is subtracted from that of the higher level (ants), positive values indicate a greater specialization of the higher trophic level (Blüthgen et al., 2007); (iv) d’ value, which indicates the variation in the specialization of the analyzed species (Blüthgen, Menzel & Blüthgen, 2006), evaluated in a range between 0 and 1, where 0 means that the species is not a specialist and 1 that the species is a perfect specialist (Dormann, Fruend & Gruber, 2019); (v) niche overlap (NO), shows the average similarity in the interaction patterns of a trophic level, it is expressed in values between 0 and 1, where the value 0 indicates that the trophic level does not share its niches and the value 1 indicates that there is perfect niche overlap (Krebs, 1989); (vi) species strength (SS), is defined as the sum of the lower trophic level dependencies on the analyzed species (Bascompte, Jordano & Olesen, 2006); (vii) interaction push-pull (PP), is the direction of the interaction asymmetry based on the number of dependencies, defining interaction asymmetry as the average mismatch between a species and dependents on the interaction effect of all species (Vázquez et al., 2007). Positive values indicate that a species affects the other level of species they interact with more strongly than they are affected by it (“pusher”). Negative values indicate that a species is, on average, being affected more in its interaction with the other species (“being pulled”) (Dormann, Gruber & Fründ, 2008).

Network metrics can be affected by confounding variables such as species richness and sampling methods (Zanata et al., 2017); one way to consider these biases is by using null model corrections (Dalsgaard et al., 2017). The Patefield null model simulates matrices where elements interact randomly without considering the degree of specialization of the network (Schleuning et al., 2014). The significance of the indices of both networks was estimated against a null model using the Patefield algorithm, generating 1,000 random matrices with a fixed distribution where the marginal totals were the sum of the interaction in both networks (Dormann et al., 2009). We calculated standardized z-scores; these were calculated using 1,000 null assembled communities, subtracting the mean of the statistic calculated across these communities from the observed value and then dividing by the standard deviation. Z values below −1.65 or above 1.65 indicate approximate statistical significance at the 5% error level (one-tailed test) (Galeano, Pastor & Iriondo, 2009). The P-value was calculated as the proportion that the observed value of the evaluated parameter was higher or lower than that of the random matrices divided by the total number of randomizations. Likewise, to test whether the values of the indices differed significantly between the oak and induced grassland networks, the same Patefield algorithm was used with 1,000 random matrices (Anjos, Dáttilo & Del-Claro, 2018). P-values and z-values were calculated in the same way as the individual parameters for each network, just that in this case, what is contrasted are the differences between the oak forest and grassland networks (observed difference vs. null model differences) (Anjos, Dáttilo & Del-Claro, 2018). Finally, the central core species of each network were identified, defining the centrality of a species based on how central or peripheral they are in the network (Dáttilo, Díaz-Castelazo & Rico-Gray, 2014). Central core species have many interactions among themselves, and peripheral species with fewer interactions interact with a proper subset of the central core of generalists (Dáttilo, Díaz-Castelazo & Rico-Gray, 2014). To find these species, the equation Gc = (Ki −Kmean)/σk was used: where Gc is the centrality of the species, Ki is the average number of links an ant species has, Kmean is the average number of links that network ants present, and σk is the standard deviation of the number of links of the network ant species (Dáttilo, Díaz-Castelazo & Rico-Gray, 2014).

Results

Ant-plant interactions mediated by resources

In the oak forest network, 11 species of ants were observed interacting with 47 species of plants in 83 interactions (Tables 2 and 3, Fig. 3A). These 11 ant species belong to eight genera, four subfamilies (Dolichoderinae, two species; Formicinae, one species; Myrmicinae, six species, Pseudomyrmecinae, two species) and four functional groups. The most represented functional group was generalized Myrmicinae (five species), followed by cold climate specialists (three species). The plant species belong to 31 identified genera and 23 plant families. Asteraceae was the best-represented plant family with eight species, followed by Fabaceae with five species. The most frequently recorded ant-plant interaction was between Prenolepis imparis Say and Perymenium mendezii DC.

Table 2 Plant species of the ant-plant interaction networks in Flor del Bosque State Park (FBST).

Oak forest (Fig. 3A) and Induced grassland (Fig. 3B). The resources or parts of the plants used by the ants are indicated: bud (bud); efn (extrafloral nectar); flo (flowers or cones); fru (fruits); hem (hemipteran honeydew); lea (leaf); by an X. Vegetation types are shown with an X.

Plants FBSP	Vegetation	Plant resources/parts visited by the ants	
Family	Genus/Species/Morpho	Codes	Oak forest	Induced Grassland	bud	efn	flo	fru	hem	lea	
Agavaceae	Agave potatorum Zucc., 1833	AGPO	X	X					X	X	
	Agave salmiana Otto ex Salm-Dyck, 1859	AGSA	X	X					X	X	
	Agave L., 1753 sp. 1	AGA1		X						X	
	Agave L., 1753 sp. 2	AGA2	X	X						X	
	Agave L., 1753 sp. 3	AGA3		X						X	
	Agave L., 1753 sp. 4	AGA4		X					X	X	
	Agave L., 1753 sp. 5	AGA5		X						X	
	Agave L., 1753 sp. 6	AGA6	X							X	
Amaryllidaceae	Sprekelia formosissima (L.) Herb., 1821	SPFO	X				X				
Anacardiaceae	Rhus standleyi F.A. Barkley, 1937	RHST	X	X	X		X			X	
Apocynaceae	Apocynaceae Juss. 1789	APOC		X			X			X	
	Metastelma angustifolium Torr, 1858	MEAN	X				X			X	
Asteraceae	Barkleyanthus salicifolius (Kunth) H.Rob. & Brettell, 1974	BASL	X	X	X X		X		X	X	
	Eupatorium deltoideum Jacq. 1798	EUDE	X	X					X	X	
	Gnaphalium L., 1754 sp.	GNAP	X						X		
	Perymenium mendezii DC., 1836	PEME	X	X			X		X	X	
	Senecio cinerarioides A. Rich, 1834	SECI	X				X			X	
	Senecio multidentatus Sch.Bip. ex Hemsl., 1881	SEMU	X		X		X		X	X	
	Senecio L., 1753	SENE		X						X	
	Stevia serrata Cav. 1797	STSE		X					X	X	
	Verbesina virgata Cav., 1795	VEVI	X							X	
	Baccharis salicifolia (Ruiz& Pav.) Pers.1807		X				X			X	
	Asteraceae 1 Bercht. & J.Presl,	AST1	X								
	Asteraceae 2	AST2	X								
Bromeliaceae	Bromelia sp.	BROM	X				X				
Cactaceae	Opuntia Mill., 1754 spp.	OPSPP	X	X	X		X	X			
Commelinaceae	Commelina L., 1753 sp. 1	COM1	X							X	
Convolvulaceae	Ipomoea L., 1753 sp. 1	IPO1	X							X	
	Ipomoea L., 1753 sp. 2	IPO2	X				X		X	X	
	Ipomoea hematica	IPHE	X				X			X	
	Ipomoea stans Cav., 1795	IPST	X								
Cyperaceae	Cyperus L., 1753 sp.	CYPE	X				X			X	
Ericaceae	Arctostaphylos pungens Kunth, 1819	ARPU	X	X			X			X	
Fabaceae	Calliandra grandiflora (L’Hér.) Benth., 1840	CAGR	X		X		X		X		
	Cologania obovata Schltdl., 1838	COOB	X				X				
	Mimosa aculeaticarpa Ortega, 1800	MIAC	X	X			X	X		X	
	Brongniartia intermedia Moric, 1836	BRIN	X							X	
	Eysenhardtia polystachia (Ortega) Sarg., 1892	EYPO		X			X			X	
	Eysenhardtia Kunth, 1824 sp.	EYSE		X			X			X	
	Fabaceae Lindl., 1836 4	FAB4	X				X				
	Fabaceae 5	FAB5		X			X				
Fagaceae	Quercus mexicana Bonpl. 1809	QUME	X							X	
Cupressaceae	Juniperus L., 1753 sp.	JUNI	X	X			X			X	
Lamiaceae	Salvia polystachya Ortega, 1798	SAPO	X				X			X	
Lilaceae	Lilaceae Juss. 1789	LILA	X				X				
Loranthaceae	Loranthaceae	LORA		X			X				
Myrtaceae	Eucalyptus L’Hér., 1788 sp.	EUCA		X						X	
Orobanchaceae	Castilleja Mutis ex L.f., 1782 sp.	CAST	X				X				
Orobanchaceae	Conopholis alpina Liebm., 1844	COAL	X				X				
Passifloraceae	Passiflora exsudans Zucc., 1837	PAEX	X	X		X					
Pinaceae	Pinus L. 1753 sp.	PINU		X						X	
Poaceae	Tripsacum dactyloides L., 1759	TRDA	X				X		X	X	
	Poaceae Barnhart, 1895	POAC		X			X				
Polygalaceae	Monnina ciliolata ex DC., 1824	MOCI	X							X	
Polygalaceae	Monnina schlechtendaliana D. Dietr. 1846	MOSC		X						X	
Rhamnaceae	Ceanothus caeruleus Lag., 1816	CECO	X						X	X	
Rosaceae	Amelanchier denticulata (Kunth) K.Koch., 1869	AMDE	X	X			X				
Rubiaceae	Bouvardia ternifolia Schltdl., 1854	BOTE		X			X				
Solanaceae	Solanum nigrescens M.Martens & Galeotti, 1845	SONI	X	X						X	
Sapindaceae	Dodonaea viscosa Jacq. 1760	DOVI		X						X	
	Morfo 4	MOR4		X			X				
	Morfo 5	MOR5		X			X				
	Morfo 6	MOR6	X				X				
	Morfo 7	MOR7	X				X				
	Morfo 8	MOR8	X				X				
	Morfo 10	MO10	X				X				
	Morfo 11	MO11	X				X				

Table 3 Ant species of the ant-plant interaction networks in Flor del Bosque State Park (FBST).

Oak forest (Fig. 3A) and induced grassland (Fig. 3B). Functional group (FG) is indicated for each ant species: CCS (Cold Climate specialists); GM (Generalized Myrmicinae); O (Opportunists); SC (Subordinate Camponitini); TCS (Tropical Climate Specialists). Ant codes used in the figures are indicated, an X indicate different types of vegetations.

Ants FBSP	Vegetation	
Subfamily	Genera/species	Codes	FG	Oak forest	Induced grassland	
Dolichoderinae	Dorymyrmex insanus Buckley, 1866	DOIN	O	X	X	
Formicinae	Linepithema dispertitum, Forel, 1885	LIDI	CCS	X	X	
	Camponotus rubrithorax Forel, 1899	CARU	SC		X	
	Nylanderia austroccidua Trager, 1984	NYAU	O		X	
	Prenolepis impairs Say, 1836	PRIM	CCS	X	X	
Myrmicinae	Crematogaster Lund, 1831 sp.	CREM	GM	X	X	
	Monomorium ebenium Forel 1891	MOEB	GM	X	X	
	Pheidole hirtula Forel, 1899	PHHI	GM	X	X	
	Pheidole nubicola (Wilson, 2003)	PHNU	GM	X	X	
	Pheidole tepicana Pergande, 1896	PHTE	GM	X	X	
	Temnothorax tricarinatus Emery, 1895	TETR	CCS	X		
	Temnothorax Mayr 1861sp.	TEMN	CCS		X	
Pseudomyrmecinae	Pseudomyrmex pallidus Smith, F., 1855	PSPA	TCS	X	X	
	Pseudomyrmex Lund, 1831 sp.	PSEU	TCS	X	X	

Figure 3 Ant-plant interaction networks in the oak forest (A) and the induced grassland (B).

Each box represents a species of plant or ant (plants orange, ants green), and the lines represent the frequency of the ant-plant interactions. The species name codes correspond to those presented in Tables 1 and 2.

The ant-plant network in the induced grassland comprises 35 plants species and 13 species of ants in 88 interactions (Tables 2 and 3, Fig. 3B). These 13 ant species belong to ten genera, four subfamilies (Dolichoderinae, two species; Formicinae, three species, Myrmicinae, six species; Pseudomyrmecinae, two species), and five functional groups. The more represented functional groups were generalized Myrmicinae (five species), followed by cold climate specialists (three species). The plant species belong to 22 identified genera and 18 plant families. Agavaceae was the more represented plant family with seven morphospecies, followed by Asteraceae with six species. The most frequently recorded ant-plant interaction was between Camponotus rubrithorax Forel and Agave potatorum. Three grassland ant-plant interactions were shared with the oak forest: Linepithema dispertitum Forel and Mimosa aculeaticarpa, Monomorium ebenium Forel with Opuntia Mill. spp. and Pseudoymyrmex Lund. spp. with Perymenium mendezii.

Resource diversity was higher in the oak forest, as more plant species were involved. The total number of resources identified for the ants was six in both vegetation types (Table 2). The more used plant resources/parts in the oak forest were floral nectar (in 41% of the plants) and leaves (in 36% of the plants). In the induced grassland, the ants also used floral nectar (32%) and leaves (46%) most. Plants with hemipterans were recorded equally in both vegetation types (oak 13%, grassland 12%). In both communities, just one plant species was registered with extrafloral nectaries (Passiflora exsudans); however, other plants could have extrafloral nectaries that are not so apparent as the “cup” type extrafloral nectaries of the Passiflora genera. Two plant species were being used as nesting sites by the ants (A. potatorum and P. mendezii) (not shown in Table 2). No significant differences were found in the ants’ use of resources in either vegetation (X2 = 1.58, d.f. = 2, P = 0.45)

Nestedness

Both in the oak forest (NODF = 48.47, NODF (Er) = 17.78, P < 0.01) and induced grassland (NODF = 60.90, NODF (Er) = 21.41, P < 0.01), the value of NODF was higher than NODF (Er), accordingly, both study sites present nestedness. The nestedness observed in the induced grassland was significantly higher than that found in the oak forest (P = 0.04). However, for WNODF, nestedness in the oak forest was significantly lower than that of the null models (WNODF = 32.74, WNODnull = 41.63, z =  − 2.57, P = 0.01). There was no significant difference between the WNODF of the induced grassland and random (WNODF = 38.14, WNODnull = 39.90, z =  − 0.3583, P = 0.48). No significant difference was found between the oak forest and induced grassland’s weighted nestedness (z =  − 0.36, P > 0.41).

Trophic and network level metrics

The oak network had an H2 value of 0.31, which was significantly higher than that expected at random (H2 = 0.16 ± 0.04, P = 0.002). The ants were more specialized than the plants (SA = 0.42), but this was not significantly different from the randomly generated matrices (SAnull = 0.39 ± 0.11, z = 0.31, P = 0.41). The niche overlap was 0.15 and 0.64 for ants and plants, respectively; both were significantly less than what would be expected by chance (antsnull = 0.29 ± 0.08, P = 0.02 and plantsnull = 0.81 ± 0.06, P < 0.002); therefore, the interaction patterns of the plants are more similar to each other than those of the ants that frequent them.

The induced grassland network had an H2 value of 0.12, which was not significantly different from the randomly generated networks (H2null = 0.10 ± 0.06, P = 0.38). Ants are more specialized in plants than vice versa (SA = 0.27), but this value was not significantly different than by chance (SA = 0.33 ± 0.13, P = 0.30). The niche overlap was 0.36 and 0.67 for ants and plants respectively; both were not significantly different than what would be expected by chance (ants = 0.54 ± 0.16, P = 0.21 and plants = 0.68 ± 0.17, P = 0.43). Although the difference in niche overlap is less than in the oak forest network, again, the interaction patterns of the grassland plants are more similar than those of the ants found on them.

Species-level metrics

Oak forest

The oak forest network is shown in Fig. 3A, and the values of the different parameters are shown in Table 4. The ant species with the highest degree and species strength was Prenolepis imparis Say, followed by Temnothorax tricarinatus Emery. With degree one, the ant with the lowest species strength was Pheidole nubicola Wilson. The plant species with the highest grade and strength was Perymenium mendezii DC. There were ten plant species with the lowest strength, all grade one. Prenolepis imparis is the ant that exerted the greatest influence (a “pusher” species) on the plants they interacted with. The ant with the least influence (a “puller” species) was P. nubicola. In the oak forest plants, P. mendezii and Eupatorium deltoideum Jacq. stand out as pushers, and the same ten plant species with the lowest strength were pullers. The most specialized plant species were Crematogaster Lund sp. and Quercus mexicana Humb. & Bonpl., which are exclusively linked to each other. Pheidole hirtula Forel is a specialist ant. The most generalist ant species was P. nubicola. The ant with the highest strength, P. imparis, was a generalist ant. For plants, the most specialized species were Sprekelia formosissima (L.) Herb. and Asteraceae sp. Seventeen plant species were the most general; these were characterized by having one link to the ant with the highest strength, P. impairs.

Table 4 Species level metrics (degree, species strength, interaction push-pull and d) for the ants and plants of the Oak forest and Induced grassland.

The highest values are shown in bold, and the lowest values for each metric are shown in italics.

Oak forest	Induced grassland	
Ant species code	Degree	Species strength	Interaction push-pull	d	Ant species code	Degree	Species strength	Interaction push-pull	d	
PRIM	39	33.29	0.83	0.15	CARU	31	23.97	0.74	0.08	
TETR	14	4.56	0.25	0.20	PSEU	19	5.28	0.23	0.10	
LIDI	15	3.98	0.20	0.40	DOIN	7	0.30	−0.10	0.04	
DOIN	4	1.68	0.17	0.68	MOEB	9	2.81	0.20	0.23	
PHHI	4	2.19	0.30	0.72	PSPA	6	0.53	−0.08	0.11	
MOEB	1	0.24	−0.76	0.67	PHTE	3	0.04	−0.32	0.05	
PSPA	1	0.01	−0.99	0.06	PHNU	2	0.02	−0.49	0.05	
PSEU	2	0.02	−0.49	0.01	CREMA	2	0.09	−0.46	0.38	
PHTE	1	0.01	−0.99	0.15	LIDI	2	0.53	−0.24	0.43	
PHNU	1	0.01	−0.99	0.00	NYAU	3	0.91	−0.03	0.66	
CREMA	1	1.00	0.00	1.00	PHHI	2	0.01	−0.49	0.04	
					PRIM	1	0.50	−0.50	0.88	
					TEMN	1	0.01	−0.99	0.27	
Plant species code	Degree	Species strength	Interaction push-pull	d	Plant species code	Degree	Species strength	Interaction push-pull	d	
PEME	6	3.04	0.34	0.08	AGPO	8	4.22	0.40	0.05	
EUDE	3	1.49	0.16	0.08	MIAC	6	1.31	0.05	0.02	
SEMU	5	1.01	0.00	0.01	BASL	6	1.59	0.10	0.02	
SAPO	3	0.39	−0.20	0.06	OPSPP	9	2.41	0.16	0.10	
MIAC	2	0.30	−0.35	0.15	AGSA	6	0.70	−0.05	0.03	
OPSPP	4	1.11	0.03	0.18	SONI	2	0.07	−0.46	0.04	
CAGR	1	0.03	−0.97	0.04	EYPO	3	0.07	−0.31	0.01	
IPO2	3	0.09	−0.30	0.00	AGA1	2	0.02	−0.49	0.00	
COAL	3	0.68	−0.11	0.38	AMDE	4	0.13	−0.22	0.05	
COM1	2	0.06	−0.47	0.02	AGA4	1	0.01	−0.99	0.02	
MOR6	1	0.02	−0.98	0.02	PEME	2	0.02	−0.49	0.00	
TRDA	1	0.02	−0.98	0.02	AGA2	1	0.01	−0.99	0.02	
SECI	2	0.26	−0.37	0.08	JUNI	1	0.01	−0.99	0.02	
ARPU	3	0.08	−0.31	0.01	AGA3	2	0.01	−0.49	0.00	
AGSA	2	0.03	−0.48	0.00	RHST	2	0.01	−0.49	0.00	
BROM	2	0.14	−0.43	0.07	PAEX	2	0.50	−0.25	0.30	
AMDE	1	0.01	−0.99	0.01	LORA	3	0.07	−0.31	0.10	
RHST	3	0.07	−0.31	0.03	POAC	3	1.01	0.00	0.47	
MEAN	2	0.10	−0.45	0.13	STSE	2	0.07	−0.47	0.13	
PAEX	1	0.01	−0.99	0.00	AGA5	1	0.00	−1.00	0.00	
IPHE	2	0.05	−0.47	0.04	BOTE	2	0.06	−0.47	0.22	
GNAP	3	0.07	−0.31	0.06	DOVI	2	0.01	−0.50	0.05	
SEPR	2	0.05	−0.48	0.09	MO12	2	0.01	−0.50	0.05	
BRIN	1	0.01	−0.99	0.00	PINU	2	0.01	−0.50	0.05	
CYPE	1	0.01	−0.99	0.00	SENE	2	0.01	−0.50	0.05	
MOR8	1	0.01	−0.99	0.00	EUDE	2	0.26	−0.37	0.65	
CECO	1	0.00	−1.00	0.00	COM2	2	0.31	−0.34	0.66	
COOB	1	0.00	−1.00	0.00	ARPU	1	0.00	−1.00	0.00	
IPO1	1	0.00	−1.00	0.00	EUCA	1	0.00	−1.00	0.00	
AGA2	2	0.05	−0.48	0.15	EYSE	1	0.00	−1.00	0.00	
MOCI	2	0.07	−0.47	0.35	FAB5	1	0.00	−1.00	0.00	
CAST	1	0.00	−1.00	0.00	MOSC	1	0.00	−1.00	0.00	
AST1	1	0.00	−1.00	0.00	MOR5	1	0.00	−1.00	0.00	
AGPO	1	0.00	−1.00	0.00	APOC	1	0.06	−0.94	0.60	
FAB4	1	0.00	−1.00	0.00	BASA	1	0.01	−0.99	0.30	
JUNI	1	0.00	−1.00	0.00						
LILA	1	0.00	−1.00	0.00						
MOR7	1	0.00	−1.00	0.00						
MO10	1	0.00	−1.00	0.00						
MO11	1	0.00	−1.00	0.00						
AGA6	1	0.05	−0.95	0.49						
QUME	1	1.00	0.00	1.00						
AST2	1	0.25	−0.75	0.77						
IPST	1	0.02	−0.98	0.37						
SONI	1	0.13	−0.88	0.66						
SPFO	1	0.25	−0.75	0.77						
VEVI	1	0.02	−0.98	0.37						

Grassland

The induced grassland network is shown in Fig. 3B, and the values of the different parameters are shown in Table 4. The ant species with the highest degree and strength was Camponotus rubrithorax Forel, while Temnothorax Mayr sp. was the species with the lowest strength and degree one. For plants, the species with the highest strength was A. potatorum, and there were five species with the lowest strength, all degree one. On the other hand, C. rubrithorax, Pseudomyrmex sp., and Monomorium ebenium were pushers, while T. tricarinatus was a puller. The plant that most influenced the trophic level of the ants was A. potatorum, and the most influenced plants were the same five species with the lowest species strength. The most specialized ant species was P. impairs, which was only found interacting with Poaceae Barnhart sp. The most generalist ant species was P. hirtula. The ant with the highest species strength, C. rubrithorax, is a generalist. The most specialized plant species were E. deltoideum and Commelina L. sp. Eleven plant species were the most generalist.

Central core species

For the oak forest network, the most central core species of ant was Prenolepis imparis (2.70), while the central core plants were P. mendezzi (3.69), E. deltoideum (1.08), Senecio multidentatus Schultz-Bpp. ex Hemsl. (2.82), and Salvia polystachia (1.8). According to the functional groups of Andersen (2000), P. imparis corresponds with a cold climate specialist. For the induced grassland, the most central core species of ants were C. rubrithorax (2.75), followed by Pseudomyrmex sp. (1.39). For plants, the central core species were A. potatorum (2.68), Mimosa aculeaticarpa Ortega (1.702), Barkleyanthus salicifolius (Kunth) H.Rob. and Brettell (1702), and Opuntia spp. (3.17). Camponotus rubrithorax and Pseudomyrmex sp. correspond respectively to subordinate Camponotini and tropical climate specialists, according to the functional groups of Andersen (2000).

Comparison of network-level metrics (Oak Forest vs. induced Grassland)

When the network-level metrics of the oak forest vegetation were compared to those of the induced grassland, no difference was found in the specialization asymmetry (z =  − 0.37, P = 0.58), nor in the niche overlap of the ants (z =  − 1.19, P = 0.87), or the plants (z =  − 0.52, P = 0.63). However, significant differences were found in the generalization level (H2); the grassland network was more generalized than the oak network (z = 3.93, P = 0.001).

Discussion

As hypothesized, the ant-plant network of the disturbed environment, induced grassland, was found to be significantly more nested (NODF) and generalist (H2) than that of the oak forest. As nestedness is related to robustness, this could mean that the grassland is more robust to species extinction than the oak forest. Differences in nestedness were more contrasting when the weighted version of NODF was evaluated; the nestedness of the oak forest was significantly lower than expected at random, suggesting a higher level of specialization than initially expected. The WNODF indicates that the grassland network is not nested, and the disturbance has led to a random network structure. There was greater specialization (SA) in the higher trophic level of both networks, i.e., the ants, and the niche overlap (NO) was low . The dominant ant species in each vegetation type may explain these differences (determined by the central core species, species strength, and degree) and their functional group, cold climate specialist P. imparis in the oak forest and the subordinate camponotini C. rubrithorax in the grassland. In fact, the central core of the oak forest network, formed by cold climate specialists, is entirely replaced by subordinate camponotini, tropical climate specialists, and generalized Myrmicinae in the induced grassland.

In accordance with other works, this study found that regardless of the type of habitat, the ants use three main plant-derived resources: floral nectar, hemipteran honeydew, the nectar from other reproductive or non-reproductive structures (leaves) and extrafloral nectar (Rico-Gray et al., 1998). However, unlike ant–plant networks in tropical environments, where the resource most used by ants is extrafloral nectar (Díaz-Castelazo et al., 2010; Dáttilo, Díaz-Castelazo & Rico-Gray, 2014; Costa et al., 2016), this study found floral nectar to be the favored resource in both vegetation types. This was an expected result, as it is known that temperate floras have a lower abundance of extrafloral nectaries than tropical regions (Coley & Aide, 1991; Morellato & Oliveira, 1991). One explanation for this difference may be related to the higher ant foraging activity in tropical vegetation and greater ant anti-herbivory efficacy of plants in temperate areas, which favors the selective advantage of EFN in the former (Morellato & Oliveira, 1991). This study shows that ant-plant networks can be structured around floral nectar and honeydew, and that the importance of these resources in temperate communities could have been overlooked. Costa et al. (2016) found differences in the nestedness, modularity, complementary specialization, and niche overlap of ant-plant networks depending on the type of resource used by the ants. Their findings highlight the importance of considering how the type of resource used by an ant community affects its network structure. The effect ants have on the plants they forage floral nectar or honeydew on is another topic to be assessed in future works.

Several studies confirm the stability of mutualistic networks in general (Bascompte & Jordano, 2007; Burgos et al., 2007; Gaiarsa & Guimarães, 2019) and ant-plant networks, in particular, remaining nested after different types of disturbances (Passmore et al., 2012; Sánchez-Galván, Díaz-Castelazo & Rico-Gray, 2012). However, according to Thébault & Fontaine (2010), disturbance would generally be expected to cause a loss of specialist species, resulting in more nested and, therefore, more generalist networks. Changes in the specialization level are apparent after disturbance (Câmara et al., 2019): increasing (Sánchez-Galván, Díaz-Castelazo & Rico-Gray, 2012; Falcão, Dáttilo & Izzo, 2015) or decreasing (Passmore et al., 2012; Emer, Venticinque & Fonseca, 2013; Lara et al., 2020), likely dependent on the turnover of ant/plant species, changes in ant/plant species roles, loss of specialist ant/plant species or even changes in ant behavior associated with different plant resource availability. Passmore et al. (2012), found that even though ant-plant mutualistic networks in the Amazonian forest are resistant to the abiotic and biotic changes caused by fragmentation, a loss of specialist species was observed in forest fragments. Emer, Venticinque & Fonseca (2013) found that the fragmentation of the forest caused by the construction of a dam in central Amazonia led to the loss of the specialist ant and myrmecophytic plant species with just one link. Corro et al. (2019), evaluated how habitat lost and habitat fragmentation affected the ant-plant networks, of a human-modified tropical rain forest in Mexico, and they found that network specialization, was favored by forest cover. In accordance with Thébault & Fontaine (2010), and similar to the works of Passmore et al. (2012), Emer, Venticinque & Fonseca (2013) and Corro et al. (2019), in the present study, we found that the change in land use from oak forest to induced grassland resulted in a loss of specialization (a more nested and generalist network in the disturbed habitat). This disturbance caused the loss of the oak forest’s central core species, a cold climate specialist, and its replacement by subordinate camponotini, tropical climate specialists, and generalized Myrmicinae in the induced grassland. Whereas, when interaction frequency was taken into account for the calculation of nestedness, an unexpected result was found: none of the networks were nested. This finding was explored by Staniczenko, Kopp & Allesina (2013), who tested 52 empirical networks and found that all but one of the networks were binary nested. However, a nested pattern was detected in just three of the quantitative networks. They explain that species preferences are partitioned to avoid competition, which can give local stability (i.e., the ability of the system to return to the equilibrium point after small perturbations). Competition is considered one of the most important structuring mechanisms of ant communities (Hölldobler & Wilson, 1990), and ants can be categorized by dominance hierarchy levels (for a review, see Cerdá, Arnan & Retana, 2013), in which dominant ants tend to exclude submissive ants. Submissive ants have different strategies to deal with the dominants, such as finding the resource before the dominant ants (discovery-dominance trade-off; Fellers, 1987), foraging during the hours of the day with more extreme temperatures (dominance and thermal tolerance trade-off, Cerdá, Arnan & Retana, 2013), or using different food resources. The non-nested pattern in the quantitative matrix found that these habits could be related to the submissive ants’ tendency to avoid competition with superior dominant ants by using the resources with different frequencies.

In the oak forest, the central core species with greater strength and pushers are cold climate specialists. While in the induced grassland, the central core species with greater strength and pushers are subordinate Camponotini, tropical climate specialists, and generalized Myrmicinae, which are found in warmer open environments (Andersen, 1997). Cold is a stressor for ants (Andersen, 2000); the change from closed woodland to open vegetation creates warmer, ergo less stressful conditions for the activity and presence of ants, which favors the displacement of specialists by a community of generalists. This result is similar to that found in a temperate site by Lara et al. (2020); the increase in temperatures resulting from the change of oak forest vegetation where cold climate species usually prevail to grassland favored the presence of ruderal plants and ants tolerant to higher temperatures. Disturbance could create more favorable conditions for certain groups of ants, especially in temperate environments (Cuautle, Vergara & Badano, 2016), because low temperature is a stressful factor for ants (Andersen, 2000). In temperate environments, disturbance could create open vegetation with warmer temperatures that can be favorable, at least for certain types of ants such as tropical climate specialists, generalized Myrmicinae and opportunistic species (Andersen, 2000; Cuautle, Vergara & Badano, 2016). Higher temperatures and solar radiation, and lower humidity were recorded at our study site in the induced grassland compared to the oak forest (Barranco-León et al., 2016). Given that land-use change produces a change in microclimatic conditions, the loss of cold climate specialists is imminent in temperate areas (Cuautle, Vergara & Badano, 2016). This outcome is similar to that reported by Benítez-Malvido et al. (2014). They compared the visitors (herbivores, omnivores, and predators) of a Heliconia network in primary forests and forest fragments, finding that the specialist herbivores of the original forest had been lost in the forest fragments.

The central core species in the oak forest network was the cold specialist P. imparis. Prenolepis imparis is a dominant ant found foraging on practically all plants in the oak forest of FBSP. This ant is characterized by its adaptation to cold environments, aggressive behavior, and ability to forage at low temperatures (Wheeler, 1930; Lynch, Balinsky & Vail, 1980; Fellers, 1989; Andersen, 1997; Juárez-Juárez et al., 2020). Prenolepis imparis depends heavily on honeydew at the study site, with a marked seasonality to its foraging; it is almost exclusively found collecting honeydew on plants in the rainy season (Cuautle per. obs.). Honeydew is a resource appreciated by ants and for which there is usually even greater competition than for extrafloral nectar (Fagundes et al., 2016). The low niche overlap between the ants, coupled with the specialization at this trophic level and the species strength of P. imparis, would seem to indicate that P. imparis dominates the plant and Hemiptera-mediated resources in the oak forest community. This species’ dominance in the network is evident when observing how the presence of P. imparis determines the distribution of ant interactions, especially on those that follow in species strength. For example, although T. tricarinatus is found on plants that P. imparis has a high intensity of interaction with, such as P. mendezii, it interacts less with these plants, choosing instead plants with which P. imparis has a lower intensity of interaction or does not interact with at all (e.g., Asteraceae 2, Verbesina virgata Cav.). The same can be observed for Linepithema dispertitum.

The induced grassland community is dominated by C. rubrithorax, Pseudormyrmex sp., and M. ebenium; that is, the community is essentially formed by subordinate Camponotini, tropical climate specialists and generalized Myrmicinae (sensu Andersen, 2000). Cold climate specialists P. imparis, Temnothorax sp., and L. dispertitum are relegated to the periphery of the network and are associated with only one or two species of plants; their presence on the edges of the grassland is possibly due to an “edge effect”. Camponotus rubrithorax is a generalist species that usually dominates resources such as extrafloral and floral nectar and honeydew; this species is also characteristic in open environments with warm temperatures (Rico-Gray et al., 1998; Guzmán-Mendoza & Castaño Meneses, 2007). Despite the considerable competition between C. rubrithorax and the other dominant species, as evidenced by the low niche overlap, C. rubrithorax and Pseudomyrmex sp. are able to coexist. For example, C. rubrithorax and Pseudomyrmex sp. frequent the same plants (A. potatorum, B. salicifolius, Opuntia spp., A. salmiana, Fig. 3B), although Pseudomyrmex sp. do so with less intensity. Pseudomyrmex sp. was found mainly on A. potatorum and was observed tending Hemipteran on it, foraging for liquids, and possibly using this plant as a nesting site (Cuautle obs. pers.). Agave potatorum, an agave endemic to Mexico (García-Mendoza, 2010), was used in the study site for the reforestation of pastures (Martínez com. pers.). Its position in the network corroborates a finding by Díaz-Castelazo et al. (2010), i.e., species introduced to native vegetation may be incorporated into the ant-plant interaction network and may even form part of the core of the network. Although plant-Hemiptera-ant interactions were also found in the grassland, the most important resource offered by its plants was floral nectar. For instance, one of the central core plants, M. aculeaticarpa, was a significant source of nectar; the same for Opuntia spp., although the presence of EFNs and defense by ants have been reported in this genus (Oliveira et al., 1999). Barkleyanthus salicifolius, the other central core plant, also contributed by offering floral nectar, but in addition, ants have been observed foraging secretions from this plant.

In the oak forest, the most generalist ant was P. nubicola; although it had just one link, that link was to a generalist plant with the greatest strength in the network, P. mendezii. The most generalist ant species in the induced grassland was P. hirtula, linked to two plant species with the highest grade and strength (A. potatorum and Opuntia spp.). Pheidole is one of the world’s most diverse ant genera. In warm climates, it can dominate in number of colonies and workers, and biomass (Wilson, 2003). However, in this study, the species of Pheidole found in both networks were not central. This finding could be related to their preference for other food resources besides nectar, mainly seeds (Whitford et al., 1981), and their higher abundance at ground level (Wilson, 2003). The most specialized ants were Crematogaster in oak forest and P. imparis in grassland. These species interact with a single plant species, the first with Q. mexicana and the second with Poaceae, which reflect the interaction’s rarity rather than species-specific interactions. This is especially true for P. imparis, a characteristic species of the oak forest network but rare in the grassland.

As for the plant interactions, many plants were designated as generalists in both the oak forest and induced grassland because they were linked to the more generalist ant species. The large number of generalist plant species may indicate the foraging activity and dominance of the central core ant species of the web, which are sometimes the only ant species reported on these plants. In the oak forest, there is a small group of specialist plants linked to the network’s central core ants (except for P. imparis). The presence of these central core ants on this group of plants is likely due to their avoiding competition with P. imparis and may contribute to the lower generalization level of the oak forest network compared to the grassland.

The change in land use from oak forest to induced grassland generated a more nested and generalized ant-plant network due to the displacement of the core of cold climate specialists in the oak forest by a core of generalist ants in the grassland. The network structure in these environments seems to be determined by the different microclimatic conditions, available plant resources, and ant dominance. Although nestedness is associated with greater resilience, in this case, the increase in the level of nestedness in disturbed vegetation can be misleading. Greater nestedness in the grassland signals the loss of the cold climate specialists characteristic of these temperate ecosystems. Furthermore, the replacement of the specialists by a more generalist community, probably more resilient to disturbance, could make it more difficult for the grassland to return to the original ant-plant interactions of the oak forest community. The loss of these interactions can affect tri-trophic interactions or the loss of the dominant ant species that keep opportunistic ant species at bay. Our results illustrate the fragility of temperate ecosystems and how increased temperatures resulting from the change in land use and climate change leave them especially vulnerable to the loss of their biodiversity.

Supplemental Information

Data S1 Matrices of plant-ant interactions in FBST

S1a. Oak forest matrix of plant-ant interactions in FBST (Flor del Bosque State Park)

S1b. Grassland matrix of plant-ant interactions in FBST (Flor del Bosque State Park)

Click here for additional data file.

Thanks go to F. Luna-Castellanos, Uri Gonzalez Díaz and María Gómez-Ortigoza for field assistance. We also thank, W. Mackay and G. R. Toledo-Pérez for the ant species identification and R. Acosta-Pérez for the plant species. We thank Wesley Dáttilo for his R scripts to estimate the significance of the network descriptors. Additional thanks to Rachel M. West for the English proofreading of this manuscript. We also thank four anonymous reviewers.

Additional Information and Declarations

Competing Interests

Author Contributions

Field Study Permissions

Data Availability

The authors declare there are no competing interests.

Mariana Cuautle conceived and designed the experiments, performed the experiments, analyzed the data, prepared figures and/or tables, authored or reviewed drafts of the article, and approved the final draft.

Cecilia Díaz-Castelazo conceived and designed the experiments, analyzed the data, authored or reviewed drafts of the article, and approved the final draft.

Citlalli Castillo-Guevara conceived and designed the experiments, analyzed the data, prepared figures and/or tables, authored or reviewed drafts of the article, and approved the final draft.

Carolina Guadalupe Torres Lagunes analyzed the data, prepared figures and/or tables, authored or reviewed drafts of the article, and approved the final draft.

The following information was supplied relating to field study approvals ({i.e.}, approving body and any reference numbers):

The Secretaria de Medio Ambiente y Recursos Naturales (SEMARNAT) (Secretary of Environment and Natural Resources) approved the collection permit SGPA/DGV/06901/15

The following information was supplied regarding data availability:

The raw data is available in the Supplemental File.

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
