# Peer review of "Changes in the core species of the ant-plant network of oak forest converted to grassland: replacement of its ant functional groups"

_PeerJ, doi:10.7717/peerj.13679_

## Round 0.1 · original submission · Major Revisions

Dear Author
Thank you for choosing PeerJ for your submission. The article needs a lot of work for consideration in this journal. Please do needful against the comments/suggestions of reviewers.

Reviewer 2 has suggested some articles for the citations update. Authors are independent to cite or not even they can find other relevant articles if any. If you do not include them, this will not influence my decision.

Best

Reviewer 1 ·

Basic reporting

Even though I am not a native English speaker, I noticed some language errors and sentences that can be improved to facilitate understanding. Therefore, I suggest an English revision of the article.

Experimental design

'no comment'

Validity of the findings

'no comment'

Additional comments

This study focused an interesting topic on ant-plant mutualistic interactions, which was not well-studied so far (in temperate ecosystems). The analyses are overall well-described with suitable methodology and interesting conclusions. In the introduction and materials and methods sections, sentences are well-organized and suitable references are provided. But the authors should overall revise the texts in the results and discussion section, including figure captions, because of awkward sentences and lack of explanation. Finally, even though I am not a native English speaker, I noticed some language errors and sentences that can be improved to facilitate understanding. Therefore, I suggest an English revision of the article.

Therefore I would suggest "Major revision" to be acceptable for the PeerJ.

I have a few minor comments which I list below:

Introdution; L118-123: Okay, I see this being explored here and at the beginning of your discussion you address the issue of functional groups, which I think is fantastic and which is something that greatly enriches the study. However, you do not address throughout the methodology that the study ants classified into functional groups, likewise this is not explored as expected in the results. Finally, I suggest you present as supplementary material this ant functional group classification.

Methodology; L131-133: I think you have to add the map of study area to this paper. Many readers of this journal does not know Mexico geography. I hope that you will be able to add the map of study area so that people all over the world can understand it.

L156-157: Haven't samples been established independently along the transects? Does every transect correspond to a large sample? I understand that for the intended analyses, this may not have significant implications, however, I believe you have to detail this information here.

L176-178: So the ants were collected, I imagine via manual collection right? Using tweezers? Entomological vacuums? This has to be made clear in your methodology. Please rephrase and add this information a few paragraphs above.

Results: I miss the biological part here. I suggest starting by exploring the ant and plant community. I would start with that part. How many interactions were most frequent throughout the study? Which were unique to each environment? You present some of this information throughout the paragraphs, but I find it difficult to follow the presentation of the results as it is, please reformulate.
Also, the part of classifying ants into functional groups that is presented throughout the discussion should be presented here. Please rephrase this.

Discussion: Discussion too extensive and partly difficult to follow. I believe that following the same objective logic as presented in the methodology and results would be fundamental here. Part of what is presented in some paragraphs can be properly presented in the results and the text reformulated here.

L334-338: As I mentioned before, I miss the presentation of the results referring to this classification of the sampled species into functional groups. Some of what you explore here throughout the discussion (in the third and fourth paragraphs) I expected to appear earlier in the results section. This will allow you to summarize some of the unnecessary information presented here.

L458-459: How many? I expected this information to be properly explored in the results section.

L459-475: I understand that part of what is presented here has already been presented earlier in the results, and that it can be discussed in another way. I suggest rephrasing and summarizing this information.

Figure 1: Use the letters (a) and (b) to distinguish the environments, and reformulate the legend.

L668-670; Figure 2 legend: Please inform here in the legend that the species name codes correspond to those presented in table 1, this is essential.

Table 1: I suggest some reformulations in this table. Present a first column with the name of the family, a second with the name of the species - which must contain the appropriate scientific name of the species with author and year -, a column with the code (use "code") and finally a column distinguishing the type of environment. And use an “X” to indicate the occurrence of the species. This will make the table more adequate and even with less unnecessary information.

Specific comments:
L29: replace “the nested structure and specialization level” with “network properties”

L87: replace “in” with “on”

L88: remove "a" from the phrase "a not well-connected groups of species"

L95: replace: “in temperate ecosystems than in tropical ones” with “in temperate than in tropical ecosystems”

L105: remove space before "Therefore"

L108: replace “work” with “study”

L329: replace “As expected, the” with “The” [I particularly don't like to start a paragraph like this, it gives little or almost no weight to the study's findings. It could be something like hypothesized, hence the whole meaning changes. However, as they preferred not to work with hypotheses, or not to leave this formulated in the study (which would easily be possible) I make this small suggestion for a change.]

L406: remove space before " Disturbance"

L406-409: Checking the English of this part, I found it confusing. I suggest rephrasing to allow proper understanding of the information.

L426: replace “ants” with “ant”

L477: replace “As expected, the” with “The”

L488: replace “The above” with “Our results”

Reviewer 2 ·

Basic reporting

This is a good manuscript have explained and fundamentally sound research. It is well written and have a coherent structure that is able to address your research in descriptive way.
Introduction:
This section is well described the importance of research as well the objectives. Background information is very clear but there is lack of hypothesis in the paper. Generally, there is need to remove a paragraph from the introduction as it looks more appropriate in the discussion to support the results, it have also mentioned in general comments section. Although citations are not latest as I have suggested some latest articles to must cite it, which will make it better. References are mentioned in the annotated attached file which will be open in adobe reader.

Experimental design

This part is well written, and each point of experiment layout has been clearly described by the author. Few comments for improvements have been given in the general comments section. Please add the map of study and it is good to add the table of climatic conditions of the studied region.

Validity of the findings

Well explained and appropriate results according to the research objectives. Discussion have needed some grammatically improvement, and also requires more citations to support the results in discussion.
Please remove figure and table numbers from the discussion.
Figures:
Figures require more improvement in description of each figure.

Additional comments

1- In line “60” change the font size of “essentially”.
2- In line “72” change “Contribute to community” to “Contribute in community”.
3- In line “87” the word “changes can have” should replace by “changes may have ”
4- In line “87” “and can even cause” should replace by “and also cause”
5- Lines# 105 to 111 should be the last lines of INTRODUCTION. Though ,it would seem more better in the end.
6- Lines 118 to 123 may use in the discussion chapter rather than in introduction. It would support the results in that chapter.

Annotated reviews are not available for download in order to protect the identity of reviewers who chose to remain anonymous.

Reviewer 3 ·

Basic reporting

The manuscript evaluated the difference in the structure of facultative ant-plant networks (metrics for network and species levels) between an Oak Forest and grassland in the same region. The authors went several times to the field to record the interactions and, though they don’t have spatial replicas, they have a good picture of ants foraging on plants at these two types of vegetation. I suggest here some points that I guess have the potential to improve the quality of the manuscript.
Authors are not consistent with the use of “plant-ant” or “ant-plant” (e.g., line 152), although they used most of the times “plant-ant”. In the literature “ant-plant” and also “animal-plant” are the most used terms. If the authors have a special reason to use plant-ant it could be justified, but if not, I strongly recommend the use of ant-plant (please check throughout the manuscript).
Here I summarize my main concerns, but they are detailed in the following pages: (i) One of my most important concerns is related to what the authors consider as a land-use change because, in my opinion, they are actually comparing two vegetation types. They argued that the grassland is recovering in the last 80 years ago, so it is not an ordinary land-use change, it is established vegetation through this time. To do the comparisons between land-use changes, the authors would need to define replicas obtaining a measure of the variation within the Oak Forest and the grassland. Even better when it is possible to proceed with the BACI (Before After Control Impact), one can tell that this is the real effect of the land-use change. In the case of the study presented here, they can only infer how the networks are structured in different vegetation types or different habitat conditions. (ii) Authors use several examples of other types of biotic interactions, not properly compared with a facultative ant-plant interaction network, like pollination, etc. I suggest focusing on ant-plant interactions and/or co-occurrences, which are a good number of publications available, and they can mention other networks, as pollination, briefly, in a couple of sentences. (iii) Also, and related with the latter, authors mentioned in the M&M that they observed and recorded each type of interaction between ants and plants but they never mentioned it in their results. I strongly recommend exploring this data, because this will improve the strength of this manuscript. (iv) Authors need to introduce and present hypotheses for each of the metrics they measured, i.e., bring the biological meaning for each of the metrics they used. In the last paragraph of the introduction, they presented basically a hypothesis related to the nested structure of networks and in the M&M and Results they present several other interesting metrics, but they have to tie them with the purposes of the work. (v) The authors mentioned a couple of times about functional groups based on a previous work of Andersen (2000) but I was not able to see which ants belong to these groups and how they evaluated it in the present manuscript. (vi) There is a lack of description of some statistical analysis they used to compare metrics, and also more details regarding the statistics of each test. (vii) The discussion needs deep improvement. I used the first two paragraphs to give some examples of the concerns that I noticed. Also, the authors need to stress each of the results they discuss, avoiding repeating the discussion of the same topic more than once. (viii) Authors argued that in grassland there is a resilient network and it is difficult to return to the interactions of the oak forest. However, the species that occupy these sites are related to the vegetation structure. So, if in the grasslands it is possible to recover the same habitat structure of the oak forest, maybe the ants that occupy these sites will also change.

1 – Title: I strongly recommend including some results/conclusions of the study in the title. My opinion is that a “comparison” is less attractive to the reader.

2 – Abstract: This section needs deep improvement.
2.1 – (Lines 24-26) The definition of nestedness in the second sentence of the abstract and with a lower connection with both prior and next sentences seems inappropriate. Besides, this definition is a bit confusing.
2.2 – (Lines 28-30) This is one of my main concerns. Change in land-use sounds like an experiment with before and after. They have three transects of samplings (pseudoreplicates) merged in the same matrix and the networks derived from it. So, they have no replicas of grasslands and oak forests, and they cannot infer about change in this sense. They have a comparison of two different environments, in terms of vegetation structure and other characteristics. Also, the history of this past intervention is not well explained (How was this intervention? See my comments for the method section). Finally, I have some concerns related to the aims/hypotheses, but please see my comments in the introduction section.
2.3 – (Lines 34-36) Based on some network metrics authors inferred about resilience and based on that they conclude that “it will be more difficult for it to return to its original state”. Is the original state related to the network structure and ant-plant interactions or the vegetation structure? The authors also used this idea in the last paragraph of the paper (Line 487).
2.4 – (Lines 36, 37) More vulnerable than what? Compared with tropical? Based on what? It is not the conclusion of this study.

3 – Keywords: This paper explores network analysis, ant-plant interactions, different vegetation structures, habitat conditions, and other subjects. I think that keywords more related to the main points of the paper should be better than the species names of some organisms studied.

4 – Introduction: Some aspects of the work were not well introduced, and some topics are disconnected from the main proposal of this study. Please see below some examples of aspects that deserve attention.
4.1 – (Line 46): Sentence without citation. I found some others sentences that the authors missed citing properly. Please check it carefully.
4.2 – (Line 49, 50): This sentence is not related to the subject of the study and also there is no citation. I recommend removing it.
4.3 – (Line 53): “land-use change” instead of “land use”?
4.4 – (Lines 54, 55): The most important for what? Maybe there is some information missing here.
4.5 – (Lines 59, 60): Sentence without citation.
4.6 – (Line 63): “has made it possible” is weird, please rephrase it.
4.7 – (Lines 64, 65): Is there a specific reason to use "plant-ant" instead of "ant-plant"? I can tell that it is widely used "ant-plant" in scientific publications.
4.8 – (Line 73): Why there is resistance to species loss? How does it work? It needs to be better explained and tied with your work.
4.9 – (Lines 80-82): How it is stated seems like it was not explored before, but several works evaluated it.
4.10 – (Lines 88, 89): And what it means or what are the consequences of it in biological terms?
4.11 – (Line 94): Cite examples or a review that explored it.
4.12 – (Lines 95-97): Sentence without citation.
4.13 – (94-101): Along with the introduction, and especially in this paragraph, authors based a lot on other mutualistic networks non-ant-plant. However, there are fundamental differences between other mutualistic networks authors mentioned and ant-plant networks. Also, as the authors have mentioned, within ant-plant interactions there are lots of variations in the predictable network when they are mutualistic or not. My suggestion is to focus on ant-plant networks, only mentioning other networks to put the results in the context of the big picture, for example, at the end of the discussion or the very beginning of the introduction.
4.14 – (107, 108): I do not agree. There are lots of papers that evaluated how ant-plant networks change with disturbances. Authors can search, for example, on Google Scholar for [ant plant disturbance] or [ant-plant anthropogenic] and they will find some publications, as a whole book dedicated to this subject (organized by Oliveira and Koptur), and some articles that investigate the fire, dam, anthropogenic, clearcutting, forest fragmentation, etc. After I made this comment, I saw that the authors cite one of these papers, my point here is as the authors stated seems like no one ever evaluated how disturbance affects ant-plant networks.
4.15 – (Line 110): Here is one of my main concerns, the aims/hypotheses. The authors chose to consider a priori that the structure of both networks would be nested. On the other side, the authors have no hypotheses/expectations for the all-other metrics they measured. If the authors analyze data, they have to introduce some information about this analysis and they have to state a biological expectation for that analysis.
4.16 – (Line 110): Seems out of context an introduction referring a lot to mutualistic networks and then the authors are studying facultative ant-plant networks. First of all, what is a “facultative ant-plant network”? In this system, plants are protected by ants? I suggest being clear about how the information regarding the other networks (mutualistic) can be used to a facultative ant-plant network and the limitations of these comparisons.
4.17 – (Line 112): Maybe “disturbance loving species” sounds weird.
4.18 – (Lines 113 & 117): Emer et al., 2017 is not in your reference list. It is very serious, I did not check all references, but I suggest the authors do it carefully.
4.19 – (Line 126): As I mentioned in my comment 2.2, the original network sounds like a before-after comparison. Here you have two different vegetation types, with no replicas.

5 – M&M: Lacks important information, needs deep improvement.
5.1 – (Lines 136-146): I miss a better description of the study area. The tall of the oak trees, the understory of the oak forest, the height of the grass, the shadow/sunny at these areas, etc. If the different vegetation types are interspersed among them or if they are in continuous patches, etc.
5.2 – (Line 169): “foraging” instead of “frequenting”?
5.3 – (Lines 161-170): Authors spent a long-time doing fieldwork and recording the ants and the resource they were foraging on. However, they do not explicitly explain which types of interactions they consider, and how they explored them (I didn’t find any information regarding types of interactions in this paper). They have to describe each of them in detail and present them in the Results. The types of interactions between ants and plants can open an interesting and novelty perspective to this system, nicely strengthening this manuscript; e.g., Costa FV, Mello MAR, Bronstein JL, Guerra TJ, Muylaert RL, Leite AC, et al. (2016) Few Ant Species Play a Central Role Linking Different Plant Resources in a Network in Rupestrian Grasslands. PLoS ONE 11(12): e0167161. https://doi.org/10.1371/journal.pone.0167161.
5.4 – (Lines 172-176): Split these ideas into two sentences, it is large and confusing.
5.5 – (Line 179): If it is in English there is no accent in Mexico. But I suggest keeping the original name of the institution in this case, keeping the name of Mexico country in its Spanish form.
5.6 – (Line 180): Here you can mention that the ants and plants were identified by specialists and include them in the acknowledgments. Also, in the acknowledgments, the authors cited again these names and in addition W. Mackay, not cited here.
5.7 – (Line 181): Please provide the data related to the curation of the specimens.
5.8 – (Line 190): “i” and “j” in italic.
5.9 – (Line 191): If authors have the weighted data, why not use the WNODF; e.g., Britton, N., Neto, M. A., & Corso, G. (2016). Which matrices show perfect nestedness or the absence of nestedness? An analytical study on the performance of NODF and WNODF. Biomath, 4(2), 1512171.?
5.10 – (Lines 198, 199): The higher the D, they are more generalist? Is it compared among them or there is a scale?
5.11 – (Lines 198-218): As I said before, there are no expectations for these analyses. In the last paragraph of the discussion, authors need to bring the reasons they are analyzing those metrics, and the biological meaning of each of them must be introduced in the introduction section.
5.12 – (Lines 231-233): The definition of generalist and specialist core species is proposed by Dáttilo, according to the appropriate equation. These other authors didn't mention the core species under the same terms.

6 – Results: First of all, authors need to present the basic results of their study, as how many species of ants and plants, belong to what subfamily and family, etc. Also, they have to present the exploratory data, as the frequency of ants and plants on each type of vegetation, etc. Also, I recommend being consistent with the use of two or three decimal degrees, for example, NODF = 48.47 (Line 242) and SA = 0.422 (Line 253).
6.1 – (Lines 244, 245): How did the authors get this P? Which statistical test is that? It is a serious concern. All statistical analyses must be presented in the data analysis section. In addition, the simple P does not tell too much about the statistical test.
6.2 – (Lines 245-247): This seems like a discussion and interpretation of the results.
6.3 – (Lines 271-318): Especially for the subsection “Species-level metrics”: Impossible to follow these results in the text, I suggest including a table or tables presenting them.
6.4 – (Line 313): Is P. imparis more central than others?
6.5 – (Lines 320-325) The same as the comment 6.1: How those comparisons were made?

7 Discussion: It needs improvement. For example, sometimes they mentioned the results in “network” terms but did not contextualize them in biological terms. Also, I found some possible misinterpretations of some results.
7.1 – (Line 331): What is the possible cause of this result? What it does mean in biological terms?
7.2 – (Lines 333, 334): Checking the results, I found that both were low Oak: 0.15 and Grassland: 0.36 (niche overlap of ants), so there is no a greater tendency of ant specialization in grassland because even if one is higher than another one, both are low (less than 0.5). Or is it another result?
7.3 – (Lines 344-348): Is this a specialist/generalist matter or is it related to the abundance of nests? How it can be controlled/explored?
7.4 – (Lines 349, 350): I did not understand why invasive species are mentioned.
7.5 – (Lines 352-355): Where are the results of your work related to different resources ants foraging on? (Comment #5.3)
7.6 – (Lines 341-364): In this paragraph, authors cited several studies not connecting them with their results; and at the end (Lines 361, 362) they re-state a result in one sentence, but lacking connection with the literature cited and the biological meaning of this result.
7.7 – (Line 366-412):
Table: The current table could bring basic information (e.g., frequency) and the metrics at the species level (e.g., d’, who is the core species, etc.), and also the codes. A table with only codes is not useful for the reader.

Figures:
Figure 1: A photo of one road is not representative of the habitat the authors did the samples – or the road was the transect?
Figure 2: Include on the caption of this figure where the reader can find the names of the species based on the codes presented.

Experimental design

I included all information in the section "Basic reporting".

Validity of the findings

I included all information in the section "Basic reporting".

Additional comments

I included all information in the section "Basic reporting".

---

## Round 0.2 · Minor Revisions

Dear Authors,

Thank you for improving the manuscript in revision. I suggest following the comments of Reviewers 2 and 3 again to make changes.

Reviewer 1 ·

Basic reporting

no comment

Experimental design

no comment

Validity of the findings

no comment

Additional comments

no comment

Reviewer 2 ·

Basic reporting

I have a suggestion in the title
"Changes in the core species of ant-plant network from an oak forest to induced grassland: replacement of ant functional groups"
I suggest to replace your title with the above mentioned because in the present title authors have used more than enough "an" which seems to be ambiguous and grammatically imperfect.

Experimental design

No comment

Validity of the findings

No comment

Additional comments

No comment

Reviewer 3 ·

Basic reporting

REVIEWER 3
I am revising the updated version of this manuscript. Now, I found an improved manuscript in general. The authors did a good job in order to address my concerns, however, I still have comments that need improvement.
One important general comment that must be clear is that the authors are not evaluating “mutualistic networks” as they are stating at the beginning of their Abstract; authors have to check the entire MS on both terminology and concept/theory/background/state of the art and change it.

I have some additional comments that I list as follow:
Comment/Response 3: I have made a search on Google Scholar for “ant plant network disturbance” and “ant plant network disturbance cooccurrence” and I have found some papers that match the purpose of the authors’ study and they can read and potentially use to cite and some papers that the authors already have cited, but can also be useful to contextualize the study:
https://doi.org/10.1111/1365-2656.12820
https://doi.org/10.1007/s10980-018-0747-4
https://doi.org/10.1007/s00114-019-1614-0
https://doi.org/10.1007/s10841-020-00216-4
https://doi.org/10.1111/cobi.12045
https://doi.org/10.1111/ens.12407
https://doi.org/10.1111/ecog.04531
My point is, avoid to using studies that did not evaluate comparable network interactions the authors are evaluating.

Comment/Response 4: As I suggested before, I strongly recommend exploring this data, in terms of data analysis and/or network analysis. Table 2 is not easy to follow, but different colors for different interactions presented on the networks or other visual/analytical approaches would be very welcome.

Comment/Response 5: I have one comment regarding the hypotheses. It is not clear why authors started mentioning “ants associated with nectar and/or hemipteran resources tend to be mutualistic, and mutualistic networks have shown to be nested”. The authors did not evaluate plants with EFNs, although very few plants had EFNs. Authors are evaluating ants foraging on the vegetation, it is definitively not a mutualistic relationship.

Comment/Response 32: I do not see the point of stressing the theory of mutualistic/facultative and not contextualizing to authors’ work. My point is to let clear to the reader what is the connection of these ideas and concepts to their target.

Comment/Response 38: Authors included the citation Costa et al. 2016 and do not cite that paper. Again, this is a very serious concern. It is not acceptable for any publication this kind of mistake. In comment 34 from my first review I highlighted that they had the same issue and they said that they “We apologize, we made another revision of the literature”; as the authors can see, it seems that the bibliography revision needs to be better done, it definitively deserves more attention.

Comment/Response 56: I completely disagree. One thing is that the network has a value different from the null model, the other thing is the value of the metric. They have completely different meanings. The value of the metric is low for both treatments as I said before, authors cannot argue that there is a greater specialization of one in relation to another if both values are lower than 0.5. Please check the meaning of the null model and what do mean being different from the null models.

Comment/Response 61: And where is the new photo?

Experimental design

Please see section 1.

Validity of the findings

Please see section 1.

Additional comments

Please see section 1.

---

## Round 0.3 · accepted · Accept

Dear Author
It is pleasure to inform you that your submission has been accepted for publication in PeerJ.

Reviewer 3 ·

Basic reporting

Now I found an improved manuscript in which the authors solved the concerns I raised. I would like to congratulate the authors that did a really great job. I hope the authors agree that they have now a strong and clear manuscript, ready to be published.

Experimental design

NA

Validity of the findings

NA

Additional comments

NA